# Permutation Picture of Graph Combinatorial Optimization Problems

## Abstract

This paper proposes a framework that formulates a wide range of graph combinatorial optimization problems using permutation-based representations. These problems include the travelling salesman problem, maximum independent set, maximum cut, and various other related problems. This work potentially opens up new avenues for algorithm design in neural combinatorial optimization, bridging the gap between discrete and continuous optimization techniques.

## 1 Introduction

The advances in Deep Learning for vision and language tasks have inspired researchers to explore its application to different domains, such as Go (Silver et al., 2016) and protein folding (Jumper et al., 2021). It has also motivated researchers to apply learning-based methods to combinatorial optimization problems (Bengio et al., 2021). This emerging field, often referred to as Machine Learning for Combinatorial Optimization (ML4CO), aims to leverage the power of machine learning to enhance or replace traditional optimization algorithms. Neural Networks (NNs) have emerged as a particularly powerful tool in ML4CO, owing to their ability to capture and process the structural information inherent in many combinatorial optimization problems. NNs can learn representations of many graph problems, making them well-suited for tasks that can be formulated as graph optimization tasks(Wilder et al., 2019; Khalil et al., 2017; Kool et al., 2018). The application of NNs in tackling graph-based optimization problems has been further exemplified in unsupervised learning fashion to a range of graph combinatorial problems, such as Travelling Salesman Problem, Maximum Clique, and other related problems (Min et al., 2024; Karalias & Loukas, 2020).

### 1.1 Supervised, Reinforcement and Unsupervised Learning

Since many graph combinatorial problems can be categorized as NP-hard, thus, direct application of Supervised Learning (SL) to these problems can be challenging due to the expense of annotation. Reinforcement Learning (RL), while it doesn't require directly labelled training data, it also has its limitations. These include sparse reward issues, where rewards can be rare, making it difficult for the agent to learn effectively, and large variance during training, which leads to unstable performance. A promising alternative approach is unsupervised learning (UL). In UL, a surrogate loss function (which can be either convex or non-convex) is constructed to encode the graph combinatorial problem. This loss function serves as a proxy for the original optimization objective. After training on this surrogate loss, the model learns a representation of the problem space. Subsequently, a decoding algorithm is applied to this learned representation to extract the final solutions. This two-step *learning followed by decoding* allows for a more flexible and potentially more effective approach to tackling graph combinatorial problems.

As shown in Figure 1, UL for graph combinatorial problems usually involves a two-phase approach. The upper row represents the *learning* phase, where a surrogate loss function is constructed from the graph problem to guide model training. The lower row is the *decoding* phase, where the learned representation is processed to construct a solution.

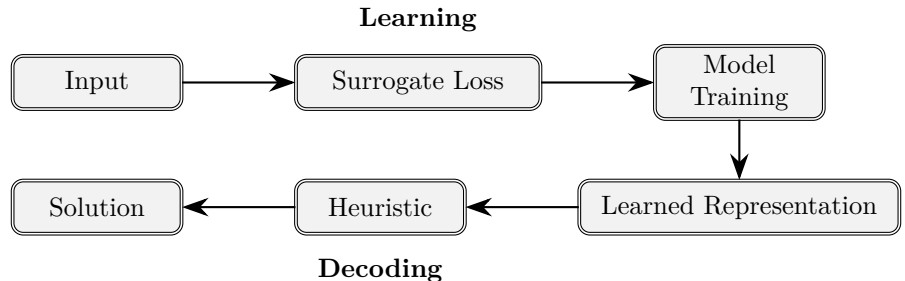

Figure 1: Unsupervised Learning Framework for Graph Combinatorial Problems

## 1.2 Building Surrogate Loss using Ising Formulation

However, building an effective surrogate loss for graph combinatorial problems is challenging. A popular choice for constructing such surrogate losses is to use the Ising model formulation derived from Quadratic Unconstrained Binary Optimization (QUBO). QUBO aligns with the binary nature of decision variables (**True** or **False**) in graph combinatorial problems. When using QUBO as a surrogate loss, the goal of machine learning is then to minimize this energy function, which corresponds to finding an optimal or near-optimal solution to the original problem (Lucas, 2014).

## 1.3 Permutation

Permutations play a crucial role in representing and solving a wide range of graph combinatorial optimization problems, offering intuitive and efficient formulations that often align closely with problem structures. In contrast, while we can use the Ising model formulations to build surrogate losses, the resulting quadratic function can have a large number of terms, leading to computational challenges. QUBO formulations essentially reduce variables to binary classifications, but the interactions between these variables may not always capture the problem structure.

In this paper, we build surrogate losses using permutations. There is a key distinction arises between our permutation picture and QUBO: we are essentially learning orders, not binary classifications. The concept of learning to order provides a more natural fit for problems where the objective is to determine a sequence or arrangement, as opposed to classification Cohen et al. (1997). This idea of learning orders has been effectively applied in other machine learning tasks, such as solving jigsaw puzzles or sorting Mena et al. (2018); Noroozi & Favaro (2016). In the following sections, we will delve deeper into how this ordering approach aligns with the structure of our problems and the potential advantages it offers over QUBO-based formulations.

The importance of formulating graph combinatorial problems as permutation operators is further emphasized in various problems. In graph isomorphism, permutations represent the bijective mappings between vertices of two graphs that preserve edge relationships, defining what it means for graphs to be isomorphic (Luks, 1982). (Babai, 2016) exemplifies the power of this approach, leveraging the theory of permutation groups to achieve a quasipolynomial time solution to the graph isomorphism problem.

The lack of intuitiveness for QUBO is more evident in problems that inherently involve sequencing or ordering, such as the Travelling Salesman Problem (TSP). In these problems, a permutation representation directly encodes the tour as a sequence of cities: $\pi = (\pi_1, \pi_2, ..., \pi_n)$, where $\pi_i$ represents the $i$-th city visited in the tour and $c_{ij}$ is the cost between city $i$ and $j$. The objective function simply becomes:

$$\text{Minimize} \quad \sum_{i=1}^{n-1} c_{\pi_i \pi_{i+1}} + c_{\pi_n \pi_1} \tag{1}$$

Here, a permutation representation is more straightforward and intuitive, directly mapping to the problem structure. Another example is the Quadratic Assignment Problem (QAP), which aims to assign a set of facilities to a set of locations, minimizing the total cost of the assignment. For a QAP with $n$ facilities and

$n$ locations, the goal is to find a permutation matrix $P \in \mathbb{R}^{n \times n}$ which minimizes:

$$\mathrm{Tr}(FPDP^T), \tag{2}$$

where $F \in \mathbb{R}^{n \times n}$ with $F_{ij}$ is the flow between facilities $i$ and $j$, $D \in \mathbb{R}^{n \times n}$ with $D_{ij}$ is the distance between locations $i$ and $j$. Overall, in many scenarios, permutations directly represent sequences, tours, or orderings, aligning closely with the structure of many problems.

## 2 Background

### 2.1 Are Discrete Variables Differentiable?

Although the discrete nature of permutations poses a challenge for the continuous operations of neural networks, several strategies have been developed to overcome this gap (Adams & Zemel, 2011; Maddison et al., 2016; Emami & Ranka, 2018; Cuturi et al., 2019; Blondel et al., 2020). These approaches can be broadly categorized into three main strategies: architecture-based solutions, gradient estimation methods, and relaxation techniques. Architecture-based solutions, such as the Pointer Network proposed by Vinyals et al. (2015), design neural network structures tailored to combinatorial problems. Gradient estimation methods, like the Gumbel-Softmax estimator introduced by Jang et al. (2016) and Maddison et al. (2016), overcome the non-differentiability of discrete operations. Relaxation techniques transform discrete permutations into continuous doubly stochastic matrices, enabling gradient-based optimization; examples include Gumbel-Sinkhorn networks (Mena et al., 2018) and the differentiable sorting method using optimal transport theory (Cuturi et al., 2019). These strategies aim to bridge the gap between discrete combinatorial problems and neural networks, facilitating the application of deep learning techniques to tasks involving discrete variables.

Notably, relaxation techniques can be seamlessly integrated with UL frameworks, allowing for end-to-end training without labelled data. This combination enables training under specific surrogate losses and learning representations that can be used to build heuristics for graph combinatorial problems.

### 2.2 Learning to Reduce the Combinatorial Search Space

Using the aforementioned techniques, researchers have developed various methods for learning the permutation operator across different tasks. These efforts extend to a wider range of problems that can be framed within the permutation picture, including graph combinatorial problems. For example, (Min & Gomes, 2023; Min et al., 2024) recently proposed an unsupervised learning method to learn the heat map of TSP by learning the permutation operator. The heat map helps reduce the search space and is then used to guide the search process.

## 3 Motivation

Since permutation-based formulations offer potential advantages over QUBO, especially for sequencing and ordering problems, and recent advancements in differentiable sorting and ranking techniques have paved the way for applying gradient-based ML methods to permutation-based formulations. In this paper, we aim to develop permutation representations for a range of (graph) combinatorial problems. Specifically, our approach focuses on building objectives by finding permutations that satisfy certain constraints. This enables end-to-end learning and hybrid algorithms that combine heuristics with learned representations. We aim to develop more flexible problem representations that integrate learning-based approaches, with a focus on creating a general framework for many graph combinatorial problems.

## 4 Formulating Graph Combinatorial Problems using Permutation

### 4.1 Travelling Salesman Problem

Let's first start from the famous TSP. TSP is a well-known NP-hard combinatorial optimization problem that has been extensively studied in the fields of computer science, operations research, and applied mathematics.

TSP asks the following question:

*Given a list of cities and the distances between each pair of cities, what is the shortest possible route that visits each city exactly once and returns to the origin city?*

TSP can be formulated as finding the shortest Hamiltonian circuit in a graph. It can also be reformulated as finding a permutation of $n$ cities that minimizes the total distance travelled. Let $\pi$ be a permutation of the set $1, \ldots, n$, where $\pi(i)$ represents the city visited in the $i$-th position of the tour. The objective is to find a permutation $\pi^* \in S_n$ such that:

$$\pi^* = \arg\min_{\pi} \sum_{i=1}^{n-1} d_{\pi(i),\pi(i+1)} + d_{\pi(n),\pi(1)}, \tag{3}$$

where $d_{ij}$ is the distance between city $i$ and $j$. Now, let $P \in \mathbb{R}^{n \times n}$ denote the permutation matrix of $\pi$, where

$$P_{ij} = \begin{cases} 1 & \text{if } \pi(i) = j, \\ 0 & \text{otherwise.} \end{cases} \tag{4}$$

For example, given the permutation $\pi$:

$$\pi = \begin{pmatrix} 1 & 2 & 3 & 4 & 5 \\ 3 & 5 & 1 & 4 & 2 \end{pmatrix}, \tag{5}$$

the permutation matrix $P$ is:

$$P = \begin{pmatrix} 0 & 0 & 1 & 0 & 0 \\ 0 & 0 & 0 & 0 & 1 \\ 1 & 0 & 0 & 0 & 0 \\ 0 & 0 & 0 & 1 & 0 \\ 0 & 1 & 0 & 0 & 0 \end{pmatrix}. \tag{6}$$

Let $C \in \mathbb{R}^{n \times n}$ denote the distance matrix with $C_{ij} = d_{ij}$. We can formulate TSP using the following matrix notation:

$$\text{Minimize}_P \quad f_{\text{TSP}}(P) = \text{Tr}(P\mathbb{V}^T P^T C), \tag{7}$$

where $\mathbb{V}_{ij} = 1$ if $j = (i \bmod n) + 1$, else 0. The objective function, $f_{\text{TSP}}(P)$, computes the total distance of the tour represented by the permutation matrix $P$. The matrix $\mathcal{H} = P\mathbb{V}P^T$ is a heat map of the tour, as noted by Min & Gomes (2023). The heat map matrix $\mathcal{H}$ can be interpreted as the probabilities of edges belonging to the optimal solution.

It is worth noting that this formulation consists of a permutation matrix $P$ and a cost matrix $C$. The permutation matrix encodes the order in which the cities are visited, while the cost matrix represents the distances between the cities. This general structure of combining a permutation matrix and a cost matrix can be used to formulate various other combinatorial optimization problems, as we will show in the following sections.

## 4.2 Maximum Independent Set

### 4.2.1 Definition

The Maximum Independent Set (MIS) problem asks to find the largest subset of vertices in an undirected graph $G = (V, E)$, where $V$ is the set of vertices and $E$ is the set of edges, such that no two vertices in the subset are adjacent to each other. In other words, the objective is to find an independent set of vertices $S \subseteq V$, where an independent set is a set of vertices in which no two vertices are connected by an edge in $E$. The problem aims to maximize the cardinality of the independent set, denoted by $|S|$. Formally, the MIS problem can be stated as: $\max_{S \subseteq V} |S|$, subject to the constraint that for all pairs of vertices $i, j \in S$, $(i, j) \notin E$.

### 4.2.2 Permutation-based objective function

We now formulate the MIS problem using a permutation-based approach similar to TSP, given a graph of $n$ vertices, the objective is:

$$\text{Maximize}_{\pi,k} \quad f_{\text{MIS}}(\pi, k) = k, \tag{8}$$

subject to:

$$\text{Tr}(PAP^T C(k)) = 0. \tag{9}$$

where $P$ is a permutation matrix corresponding to the permutation $\pi$ of the vertices, $k$ is the size of the independent set. $A \in \mathbb{R}^{n \times n}$ is the adjacency matrix of the graph, with $A_{ij} = 1$ if vertices $i$ and $j$ are connected by an edge, and 0 otherwise. $C(k) \in \mathbb{R}^{n \times n}$ is a truncation (cost) matrix defined as:

$$C(k)_{ij} = \begin{cases} 1 & \text{if } i \leq k \text{ and } j \leq k, \\ 0 & \text{otherwise.} \end{cases} \tag{10}$$

In other words, $C(k)$ is a block matrix with a $k \times k$ block of ones in the upper-left corner and zeros elsewhere:

$$C(k) = \begin{pmatrix} \mathbf{1}_{k \times k} & \mathbf{0}_{k \times (n-k)} \\ \mathbf{0}_{(n-k) \times k} & \mathbf{0}_{(n-k) \times (n-k)} \end{pmatrix} \tag{11}$$

The objective is to find a permutation $\pi$ of the vertices and a value $k$ that maximizes the size of the independent set. The constraint in Equation 9 ensures that no two vertices in the selected subset of size $k$ are connected by an edge.

### 4.2.3 Proof

Consider the matrix product $PAP^T$. This product has the property that $(PAP^T)_{ij} = 1$ if and only if vertices $\pi(i)$ and $\pi(j)$ are connected by an edge in the graph. Now consider the matrix product $PAP^T C(k)$, which correspond to the first $k$ vertices in the permutation $\pi$. The trace of a square matrix is the sum of its diagonal elements. Therefore, $\text{Tr}(PAP^T C(k))$ is the sum of the diagonal elements of $PAP^T C(k)$, which correspond to the edges between the first $k$ vertices in the permutation $\pi$. If $\text{Tr}(PAP^T C(k)) = 0$, then there are no edges between the first $k$ vertices in the permutation $\pi$. This means that these $k$ vertices form an independent set.

The objective function $f_{\text{MIS}}(P, k) = k$ maximizes the size of the independent set. Therefore, the permutation-based formulation correctly finds the MIS of the graph by maximizing $k$ subject to the constraint $\text{Tr}(PAP^T C(k)) = 0$, which ensures that the first $k$ vertices in the permutation $\pi$ form an independent set.

## 4.3 Maximum Cut

### 4.3.1 Definition

The Maximum Cut (MC) problem involves finding a partition of the vertices of an undirected graph $G = (V, E)$ into two disjoint subsets $S$ and $V \setminus S$, such that the number of edges between the two subsets, known as the cut size, is maximized. Here, we assume every edge in $E$ has the same unit weight. In other words, the objective is to find a cut $(S, V \setminus S)$ that maximizes the sum of the numbers of the edges crossing the cut, where the weight of an edge $(i, j) \in E$ is given by $e_{ij}$. The problem can be formally stated as: $\max_{S \subset V} \sum_{i \in S, j \in V \setminus S} e_{ij}$.

### 4.3.2 Permutation-based objective function

Given a graph of $n$ vertices, the objective of MC is:

$$\text{Maximize}_{\pi,k} \quad f_{\text{MC}}(\pi, k) = \frac{1}{2} \text{Tr}(PAP^T C(k)), \tag{12}$$

where $k \in \{1, 2, \ldots, n-1\}$, $P$ is a permutation matrix corresponding to the permutation $\pi$, $A$ is the adjacency matrix of the graph, $n$ is the number of vertices.

The cost matrix for $C(k)$ for Max Cut is:

$$C(k)_{ij} = \begin{cases} 1 & \text{if } (i \leq k < j) \text{ or } (j \leq k < i), \\ 0 & \text{otherwise.} \end{cases} \tag{13}$$

### 4.3.3 Proof

To prove that the function $f_{\text{MC}}(\pi, k) = \frac{1}{2}\text{Tr}(PAP^T C(k))$ correctly represents the MC problem. Let $\pi$ be a permutation of the vertices, represented by the permutation matrix $P$, and $k$ be a cut point where $1 \leq k < n$. We define two sets: $S = \{\pi(1), \ldots, \pi(k)\}$ and $T = \{\pi(k+1), \ldots, \pi(n)\}$. The matrix $PAP^T$ represents the adjacency matrix of $G$ with vertices reordered according to $\pi$, while $C(k)$ is constructed such that $C(k)_{ij} = 1$ if and only if i $\leq k < j$ or $j \leq k < i$.

The trace calculation $\text{Tr}(PAP^T C(k))$ can be expanded as the sum of two terms: $\sum_{i=1}^{k} \sum_{j=k+1}^{n} (PAP^T)_{ij}$ and $\sum_{i=k+1}^{n} \sum_{j=1}^{k} (PAP^T)_{ij}$. The first term counts edges from vertices in positions 1 to $k$ to vertices in positions $k + 1$ to $n$, while the second term counts edges in the opposite direction. Together, these sums count each edge crossing the cut exactly twice. This double counting aligns perfectly with the definition of a cut in graph theory: an edge $(u, v) \in E$ is in the cut if and only if $(u \in S$ and $v \in T)$ or $(u \in T$ and $v \in S)$.

To correct for this double counting, we multiply the trace by $1/2$ in our objective function. Thus, $f_{\text{MC}}(\pi, k) = \frac{1}{2}\text{Tr}(PAP^T C(k))$ accurately counts the number of edges in the cut defined by permutation $\pi$ and cut point $k$. It follows that maximizing $f_{\text{MC}}(\pi, k)$ over all permutations $\pi$ and cut points $k$ is equivalent to finding the MCs in $G$. This is because every possible cut in $G$ can be represented by some permutation $\pi$ and cut point $k$, and our function correctly counts the size of each such cut. Therefore, the maximum value of $f_{\text{MC}}(\pi, k)$ must correspond to the size of the maximum cut in $G$, completing our proof and demonstrating the validity of our permutation-based formulation for the Max Cut problem.

## 4.4 Graph Coloring Problem

### 4.4.1 Definition

The goal of graph coloring problem is to assign colors to the vertices of an undirected graph $G = (V, E)$, such that no two adjacent vertices have the same color. The objective is to find a valid coloring of the graph using the minimum number of colors possible, known as the chromatic number of the graph, denoted by $\chi(G)$. Each vertex must be assigned exactly one color from the set $1, 2, \ldots, k$, where $k$ is the number of available colors, and if two vertices $i$ and $j$ are connected by an edge $(i, j) \in E$, then they must be assigned different colors.

### 4.4.2 Permutation-based objective function

The problem can be written as:

$$\text{Minimize}_{\pi,k} \quad f_{\text{Coloring}}(\pi, k) = k, \tag{14}$$

subject to:

$$\text{Tr}(PAP^T C(k)) = 0, \tag{15}$$

where $k \in \{1, 2, \ldots, n\}$, $P$ is a permutation matrix corresponding to the permutation $\pi$, $A \in \mathbb{R}^{n \times n}$ is the adjacency matrix. $C(k) \in \mathbb{R}^{n \times n}$ is a block diagonal matrix, where $n$ is the number of nodes in the graph. For $k$ colors, $C(k)$ consists of $k$ blocks along the diagonal. The sizes of these blocks can vary depending on how the $n$ nodes are distributed among the $k$ colors.

Let $n_i$ denote the number of nodes assigned to the $i$-th color, where $1 \leq i \leq k$. Then the $i$-th block on the diagonal of $C(k)$ is of size $n_i \times n_i$ with $\sum_{i=1}^{k} n_i = n$. Each block in $C(k)$ is filled with 1's, and the rest of the matrix is filled with 0's.

### 4.4.3 Proof

The objective of the graph coloring problem is to minimize the number of colors used to assign a color to each vertex such that no two adjacent vertices share the same color. The objective function in Equation 14, ensures that we minimize the number of colors. The constraint in Equation 15 enforces that no two adjacent vertices share the same color. Here, $P$ is a permutation matrix, and $A$ is the adjacency matrix that encodes the edges between vertices. The cost matrix $C(k)$ assigns a value of 1 if two vertices share the same color and 0 otherwise. Therefore, the product $PAP^T C(k)$ checks whether any adjacent vertices are assigned the same color. If adjacent vertices share a color, the constraint is violated, and if no conflicts exist, the constraint holds. Hence, this formulation correctly models the graph coloring problem.

## 4.5 Minimum Vertex Cover

### 4.5.1 Definition

The goal of Minimum Vertex Cover (MVC) problem is to find the smallest subset of vertices in an undirected graph $G = (V, E)$, such that every edge in the graph is incident to at least one vertex in the subset. In other words, the objective is to find a vertex cover of the graph, denoted by $S \subseteq V$, where a vertex cover is a set of vertices such that every edge in $E$ has at least one of its endpoints in $S$. The problem aims to minimize the cardinality of the vertex cover, denoted by $|S|$. Formally, the MVC problem can be stated as: $\min_{S \subseteq V} |S|$, subject to the constraint that for all edges $(i, j) \in E$, either $i \in S$ or $j \in S$ (or both).

### 4.5.2 Permutation-based objective function

Given a graph of $n$ vertices, the objective of MVC is:

$$\text{Minimize}_{\pi,k} \quad f_{\text{MVC}}(\pi, k) = k, \tag{16}$$

subject to

$$\text{Tr}(PAP^T C(k)) = 0, \tag{17}$$

where $k \in \{1, 2, \ldots, n\}$, $P$ is a permutation matrix corresponding to the permutation $\pi$, $A \in \mathbb{R}^{n \times n}$ is the adjacency matrix. The cost matrix $C(k)$ can be defined as:

$$C(k)_{ij} = \begin{cases} 1 & \text{if } i > k \text{ and } j > k, \\ 0 & \text{otherwise.} \end{cases} \tag{18}$$

### 4.5.3 Proof

We prove that minimizing $k$ subject to $\text{Tr}(PAP^T C(k)) = 0$ is equivalent to finding a MVC of $G$. Consider a permutation $\pi$ of $V$, represented by matrix $P$. The matrix $C(k)$ is defined as $C(k)_{ij} = 1$ if $i > k$ and $j > k$, and 0 otherwise. Thus, $PAP^T C(k)$ represents the subgraph of the permuted graph induced by vertices $k+1$ to $n$. The condition $\text{Tr}(PAP^T C(k)) = 0$ means that there are no edges in the permuted graph where both endpoints are beyond the first $k$ vertices. Equivalently, every edge in $G$ has at least one endpoint among the first $k$ vertices in the permutation $\pi$. Therefore, these $k$ vertices form a vertex cover. By minimizing $k$, we find the smallest such cover. Conversely, any vertex cover of size $k$ can be represented by a permutation $\pi$ where the cover vertices are the first $k$ elements, satisfying $\text{Tr}(PAP^T C(k)) = 0$.

## 4.6 Minimum Dominating Set

### 4.6.1 Definition

A Minimum Dominating Set (MDS) in a graph $G = (V, E)$ is a subset $S$ of $V$ such that:

1. Every vertex not in $S$ is adjacent to at least one vertex in $S$.

2. $S$ is of minimum size among all dominating sets of $G$.

In other words, it's the smallest set of vertices in a graph such that every vertex in the graph is either in the set or adjacent to a vertex in the set.

### 4.6.2 Permutation-based objective function

We can formulate the MDS problem using the permutation-based approach:

$$\text{Minimize}_{\pi,k} \quad f_{\text{MDS}}(\pi, k) = k, \tag{19}$$

subject to

$$P(A + I)P^T \mathbf{1}_k \geq \mathbf{1}, \tag{20}$$

where $k \in \{1, 2, \ldots, n\}$, $P$ is a permutation matrix corresponding to the permutation $\pi$, $A$ is the adjacency matrix of the graph, $I \in \mathbb{R}^{n \times n}$ is the identity matrix, $\mathbf{1}_k = (\underbrace{1, \ldots, 1}_{k \text{ times}}, \underbrace{0, \ldots, 0}_{n-k \text{ times}})^T$ and $\mathbf{1}$ is the all-ones vector.

### 4.6.3 Proof

We prove that minimizing $k$ subject to $P(A + I)P^T \mathbf{1}_k \geq \mathbf{1}$ is equivalent to finding the MDS of $G$. Consider a permutation $\pi$ of $V$, represented by matrix $P$. The product $P(A + I)P^T \mathbf{1}_k$ calculates, for each vertex, the number of neighbors (including itself) that are in the set $D$, and the constraint $P(A + I)P^T \mathbf{1}_k \geq \mathbf{1}$ ensures that all vertices are included. Minimizing $k$ corresponds to finding the smallest such set, which is exactly the objective of the MDS problem. Therefore, the formulation is a valid representation of the MDS problem.

## 4.7 Maximum Clique Problem

### 4.7.1 Definition

Let $G = (V, E)$ be an undirected graph, a clique is a subset of vertices $S \subseteq V$ such that every pair of vertices in $S$ is connected by an edge in $E$. In other words, a clique is a complete subgraph of $G$ where all possible edges between its vertices are present. The objective of the Maximum Clique (MClique) problem is to find a clique of maximum cardinality, denoted by $\omega(G)$. Formally, the problem can be stated as: $\max_{S \subseteq V} |S|$, subject to the constraint that for all pairs of vertices $i, j \in S$, $(i, j) \in E$.

### 4.7.2 Permutation-based objective function

we can formulate the MClique problem using the permutation-based approach:

$$\text{Maximize}_{\pi,k} \quad f_{\text{Clique}}(\pi, k) = k, \tag{21}$$

subject to:

$$\text{Tr}(P(J - A - I)P^T C(k)) = 0 \tag{22}$$

where $J \in \mathbb{R}^{n \times n}$ is the all-ones matrix, $P$ is a permutation matrix corresponding to the permutation $\pi$, $A$ is the adjacency matrix and $I$ is the identity matrix.

The cost matrix $C(k)$ is same as the MIS problem, where:

$$C(k)_{ij} = \begin{cases} 1 & \text{if } i \leq k \text{ and } j \leq k, \\ 0 & \text{otherwise.} \end{cases} \tag{23}$$

### 4.7.3 Proof

We prove that maximizing $k$ subject to $\text{Tr}(P(J - A - I)P^T C(k)) = 0$ is equivalent to finding a maximum clique in $G$. Consider a permutation $\pi$ of $V$, represented by matrix $P$. The matrix $J - A - I$ has entries of 1 where there are no edges in $G$ (excluding self-loops), and 0 elsewhere. $C(k)$ is a matrix with ones in the upper-left $k \times k$ block and zeros elsewhere, effectively selecting the subgraph induced by the first $k$ vertices

in the permutation. The product $P(J - A - I)P^T$ represents the non-edge matrix of $G$ after reordering vertices according to $\pi$. When multiplied element-wise with $C(k)$, it selects the non-edges within the first $k$ vertices of this permutation. The condition $\mathrm{Tr}(P(J - A - I)P^T C(k)) = 0$ implies there are no non-edges among the first $k$ vertices in the permutation $\pi$, in other words, these $k$ vertices form a clique.

By maximizing $k$ subject to this constraint, we find the largest set of vertices that forms a clique, which is the definition of a maximum clique. Conversely, for any clique of size $k$ in $G$, we can construct a permutation $\pi$ that places the clique vertices in the first $k$ positions, satisfying $\mathrm{Tr}(P(J - A - I)P^T C(k)) = 0$. Therefore, the optimal solution to this formulation corresponds to a MClique in $G$.

### 4.8 Graph Isomorphism

#### 4.8.1 Definition

Given two graphs $G_1(V_1, E_1)$ and $G_2(V_2, E_2)$ with $|V_1| = |V_2| = n$, we want to determine if there exists a bijection $f : V_1 \to V_2$ such that $(u, v) \in E_1$ if and only if $(f(u), f(v)) \in E_2$.

#### 4.8.2 Permutation-based objective function

we can formulate the Graph Isomorphism (GI) Problem using the permutation-based approach:

$$\text{Minimize}_P \quad f_{\text{Isomorphism}}(P) = \|PA_1 P^T - A_2\|_F^2, \tag{24}$$

where $P$ is a permutation matrix corresponding to the permutation $\pi$, $A_1$, $A_2$ is the adjacency matrix of $G_1$ and $G_2$.

#### 4.8.3 Proof

This formulation directly corresponds to the definition of graph isomorphism.

### 4.9 Boolean Satisfiability

#### 4.9.1 Definition

The Boolean Satisfiability Problem (SAT) is a fundamental problem in computer science and mathematical logic that plays a central role in computational complexity theory. SAT asks whether a given Boolean formula, typically expressed in Conjunctive Normal Form (CNF), can be satisfied by assigning truth values (True or False) to its variables. A CNF formula consists of a conjunction (AND) of clauses, where each clause is a disjunction (OR) of literals (variables or their negations).

Let $X = x_1, x_2, ..., x_n$ be a set of $n$ Boolean variables. A SAT instance with $m$ clauses can be represented as:

$$\phi = \mathcal{C}_1 \wedge \mathcal{C}_2 \wedge ... \wedge \mathcal{C}_m, \tag{25}$$

where each clause $\mathcal{C}_i$ is a disjunction of literals:

$$\mathcal{C}_i = (l_{i1} \vee l_{i2} \vee ... \vee l_{ik_i}). \tag{26}$$

Here, each literal $l_{ij}$ is either a variable $x_k$ or its negation $\neg x_k$, and $k_i$ is the number of literals in clause $\mathcal{C}_i$. The SAT problem asks if there exists an assignment $\alpha : X \to \{0, 1\}$ such that:

$$\alpha(\phi) = 1, \tag{27}$$

or equivalently:

$$\bigwedge_{i=1}^{m} \bigvee_{j=1}^{k_i} l_{ij}(\alpha) = 1, \tag{28}$$

where $l_{ij}(\alpha)$ is the evaluation of literal $l_{ij}$ under the assignment $\alpha$. In this formulation: $\wedge$ represents logical AND, $\vee$ represents logical OR, $\neg$ represents logical NOT 1 represents True, and 0 represents False. The goal is to find an assignment $\alpha$ that satisfies this equation, or to prove that no such assignment exists.

### 4.9.2 Permutation-based objective function

We now encode SAT as a graph problem and then apply a permutation-based approach. We first create a vertex for each literal ($x$ and $\neg x$ for each variable $x$ in $X$). For a SAT problem with $n$ variables and $m$ clauses, our graph will have $N = 2n + m$ vertices.

Our objective is to find a permutation matrix $P \in \mathbb{R}^{2n+m}$, subject to:

$$\text{Tr}\left(CPAP^T C\right) = 0, \tag{29}$$

which is the complementarity constraint which ensures that no two complementary literals are both assigned **True** by verifying that there are no conflicting edges.

$$TPAP^T C\mathbf{1}_N \geq \mathbf{1}_m. \tag{30}$$

Here, Equation 30 is the clause satisfaction constraint which ensures that each clause has at least one literal assigned **True**, where $\mathbf{1}_N$ is a vector of ones of length $N$ and $\mathbf{1}_m = (\underbrace{0,\ldots,0}_{N-m \text{ times}}, \underbrace{1,\ldots,1}_{m \text{ times}})^T$. The matrix $A$, referred to as the *adjacency* matrix, is defined as follows:

$$A = \begin{pmatrix} V & B^T \\ B & \mathbf{0}_{m \times m} \end{pmatrix}, \tag{31}$$

where $V \in \mathbb{R}^{2n \times 2n}$ represents conflicts between literals (edges between complementary literals) and $\mathbf{0}_{m \times m}$ is zero matrix of size $m \times m$. $V$ is defined as:

$$V_{i,j} = \begin{cases} 1, & \text{if literal } i \text{ and literal } j \text{ are complementary,} \\ 0, & \text{otherwise.} \end{cases} \tag{32}$$

$B \in \mathbb{R}^{m \times 2n}$ represents edges from clauses to literals, where

$$B_{c,i} = \begin{cases} 1, & \text{if literal } i \text{ is in clause } \mathcal{C}_c \\ 0, & \text{otherwise} \end{cases} \tag{33}$$

$T$ and $C$ are two selection matrices, where they select the first $n$ positions corresponding to literals assigned **True** and ensure that each clause has at least one literal assigned **True**.

$$T = \begin{pmatrix} \mathbf{0}_{2n \times 2n} & \mathbf{0}_{2n \times k} \\ \mathbf{0}_{k \times 2n} & I_{m \times m} \end{pmatrix}, \tag{34}$$

$$C = \begin{pmatrix} I_{n \times n} & \mathbf{0}_{n \times (N-n)} \\ \mathbf{0}_{(N-n) \times n} & \mathbf{0}_{(N-n) \times (N-n)} \end{pmatrix}. \tag{35}$$

To obtain the truth assignment for the variables in the SAT problem from the permutation matrix $P$, we first extract the permutation $\pi \in S_N$, which represents the order of vertices in the graph representation of the SAT problem. The first $n$ elements of $\pi$ correspond to the literals that are assigned a value of **True** in the satisfying assignment. Each of these elements represents either a positive literal (variable) or a negative literal (negated variable).

### 4.9.3 Proof

We begin by considering a satisfying assignment $\alpha : X \rightarrow \{0, 1\}$, where

$$\alpha(x_i) = \begin{cases} 1, & \text{if } x_i \text{ is assigned } \textbf{True}, \\ 0, & \text{if } x_i \text{ is assigned } \textbf{False}. \end{cases}$$

Under this assignment $\alpha$, all clauses $\mathcal{C}_j$ evaluate to **True**. To represent this assignment in our permutation-based framework, we construct a permutation $\pi$ such that:

1. The first $n$ positions correspond to literals assigned **True**.

2. The next $n$ positions correspond to the remaining literals.

3. The last $m$ positions correspond to the clauses $\mathcal{C}_j$.

Formally, for each $i = 1$ to $n$, we define $\pi(i)$ as follows:

1. If $\alpha(x_i) = 1$, we place the literal $x_i$ at position $i$.

2. If $\alpha(x_i) = 0$, we place the literal $\neg x_i$ at position $i$.

The order of literals in positions $n+1$ to $2n$ can be arbitrary, and the clauses are placed at positions $2n+1$ to $2n + m$.

Since the assignment $\alpha$ does not assign both a literal and its complement to **True**, there are no conflicts among the literals assigned **True**. Therefore, the complementarity constraint $\text{Tr}\left(CPAP^TC\right) = 0$ is satisfied.

Moreover, each clause $\mathcal{C}_j$ contains at least one literal that is assigned **True** under $\alpha$. This implies that in the permuted adjacency matrix $PAP^T$, there is at least one edge from an assigned literal (in the first $n$ positions) to the clause vertex corresponding to $\mathcal{C}_j$. Consequently, the clause satisfaction constraint $TPAP^TC\mathbf{1}_N \geq \mathbf{1}_m$ is also satisfied.

Thus, the permutation matrix $P$ constructed from the satisfying assignment $\alpha$ satisfies all the required constraints of the permutation-based formulation. Therefore, every satisfying assignment corresponds to a permutation matrix $P$ that meets the permutation-based constraints, we provide an example in Appendix A.

## 5 Discussion

The key of constructing the permutation framework for many graph problems lies in reordering (or relabelling) the vertices. Let's take the MIS problem in Section 4.2 as an illustrative example. Recall that an independent set in a graph is a set of vertices where no two vertices are adjacent. The MIS is the largest such set for a given graph.

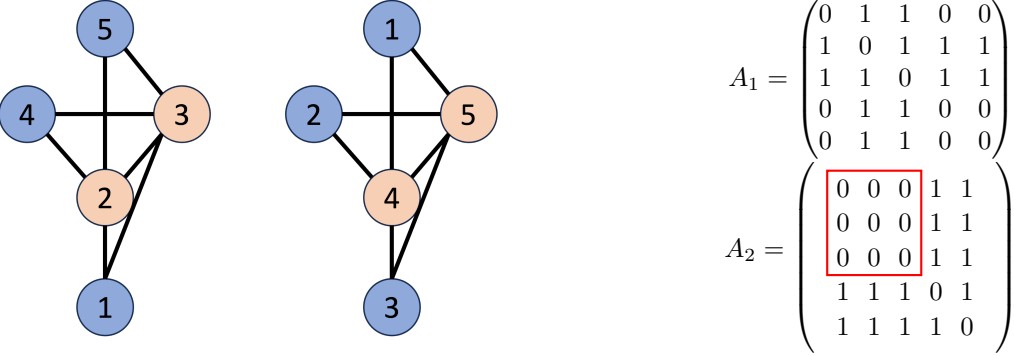

Figure 2: Comparison of graph labellings and their adjacency matrices. The blue nodes are the solution of the MIS Problem. $A_1$ is the adjacency matrix of the left labelling and $A_2$ corresponds to the right one.

In Figure 2, we see two representations of the same graph, $A_1$ shows the adjacency matrix with an arbitrary labeling of vertices and $A_2$ shows the adjacency matrix after relabelling.

The red box in $A_2$ highlights a $3 \times 3$ submatrix of zeros in the top-left corner. This submatrix corresponds to a MIS of size 3 in the original graph. This relabelling approach transforms the MIS problem into a matrix permutation problem.

Moreover, this relabelling strategy extends to other graph problems, such as clique finding and vertex cover. By framing these problems in terms of permutations, we may open up new possibilities for combining

these representations with recent advancements in learning-to-sort or learning-to-permute models. These permutation-based formulations, when integrated with these learning-based approaches, creates a framework that can potentially enhance the efficiency of solving graph combinatorial problems using neural networks.

## 6 Conclusion

This paper presents a general framework for addressing a wide range of graph combinatorial optimization problems under permutation picture. We have demonstrated the versatility of this approach by applying it to many graph combinatorial problems. It should be emphasized that there are numerous problems that can be formulated using permutation-based representations, and this paper discusses only a select few examples to illustrate the framework's broad applicability. By formulating different problems into a permutation-based structure, we provide a unified approach to tackling graph combinatorial optimization problems. For many problems, particularly those involving sequencing or ordering, permutation-based formulations offer more intuitive and direct representations of problem structures compared to traditional approaches like QUBO.

Furthermore, our approach can be combined with recent advancements in differentiable sorting and ranking techniques, opening up possibilities for applying gradient-based (differentiable) machine learning methods to these discrete optimization problems. The permutation-based framework allows for easy integration of problem-specific constraints and objectives. By transforming problems into matrix operations, our approach may be well-suited for parallel processing and hardware acceleration, potentially leading to more efficient solutions for large-scale instances. Future research directions include developing specialized neural networks architectures for learning permutations, exploring hybrid algorithms combining traditional heuristics with learning-based approaches, and applying this framework to more real-world optimization problems in various domains.

In conclusion, our permutation-based formulations offer a new perspective on neural graph combinatorial optimization problems, bridging the gap between discrete and continuous optimization techniques and leveraging ML for these problems. We anticipate that this approach will play an increasingly important role in advancing both supervised and unsupervised neural combinatorial optimization, opening up new avenues for algorithm design.

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

# A   Appendix

## A.1   SAT Example

### A.1.1   Variables and Literals

The SAT problem consists of 4 Boolean variables $x_1, x_2, x_3, x_4$ and 5 clauses in CNF:

$$
\begin{aligned}
\mathcal{C}_1 &= x_1 \lor x_2 \lor \neg x_3, \\
\mathcal{C}_2 &= \neg x_1 \lor x_3 \lor x_4, \\
\mathcal{C}_3 &= x_2 \lor \neg x_4, \\
\mathcal{C}_4 &= \neg x_2 \lor \neg x_3 \lor x_4, \\
\mathcal{C}_5 &= x_1 \lor \neg x_4.
\end{aligned}
\tag{36}
$$

We can encode the SAT problem using the permutation-based method. We first construct a graph with vertices representing literals and clauses. There are 8 total literal vertices $(x_1, \neg x_1, x_2, \neg x_2, x_3, \neg x_3, x_4, \neg x_4)$ and 5 clause vertices $(\mathcal{C}_1, \mathcal{C}_2, \mathcal{C}_3, \mathcal{C}_4, \mathcal{C}_5)$. Total number of vertices $N = 13$.

The adjacency matrix $A \in \mathbb{R}^{13 \times 13}$ is constructed as follows:

$$A = \begin{pmatrix} V & B^\top \\ B & \mathbf{0}_{5 \times 5,} \end{pmatrix}, \tag{37}$$

where $V \in \mathbb{R}^{8 \times 8}$ represents the conflicts between literals (edges between complementary literals), $B \in \mathbb{R}^{5 \times 8}$ denote the edges from clauses to literals.

Given the mapping, the complementary pairs are:

$$\begin{aligned} x_1 \text{ and } \neg x_1 & \quad (1 \text{ and } 2), \\ x_2 \text{ and } \neg x_2 & \quad (3 \text{ and } 4), \\ x_3 \text{ and } \neg x_3 & \quad (5 \text{ and } 6), \\ x_4 \text{ and } \neg x_4 & \quad (7 \text{ and } 8). \end{aligned} \tag{38}$$

Thus, the conflict matrix $V$ is:

$$V = \begin{pmatrix} 0 & 1 & 0 & 0 & 0 & 0 & 0 & 0 \\ 1 & 0 & 0 & 0 & 0 & 0 & 0 & 0 \\ 0 & 0 & 0 & 1 & 0 & 0 & 0 & 0 \\ 0 & 0 & 1 & 0 & 0 & 0 & 0 & 0 \\ 0 & 0 & 0 & 0 & 0 & 1 & 0 & 0 \\ 0 & 0 & 0 & 0 & 1 & 0 & 0 & 0 \\ 0 & 0 & 0 & 0 & 0 & 0 & 0 & 1 \\ 0 & 0 & 0 & 0 & 0 & 0 & 1 & 0 \end{pmatrix}. \tag{39}$$

We define $B$ such that:

$$B_{c,i} = \begin{cases} 1, & \text{if literal } i \text{ is in clause } \mathcal{C}_c, \\ 0, & \text{otherwise.} \end{cases} \tag{40}$$

Given the clauses and the mapping of literals to indices, $B$ is constructed as follows:

- For $\mathcal{C}_1$, literals $x_1(1), x_2(3), \neg x_3(6)$,

- For $\mathcal{C}_2$, literals $\neg x_1(2), x_3(5), x_4(7)$,

- For $\mathcal{C}_3$, literals $x_2(3), \neg x_4(8)$,

- For $\mathcal{C}_4$, literals $\neg x_2(4), \neg x_3(6), x_4(7)$,

- For $\mathcal{C}_5$, literals $x_1(1), \neg x_4(8)$.

Therefore, $B$ is:

$$B = \begin{pmatrix} 1 & 0 & 1 & 0 & 0 & 1 & 0 & 0 \\ 0 & 1 & 0 & 0 & 1 & 0 & 1 & 0 \\ 0 & 0 & 1 & 0 & 0 & 0 & 0 & 1 \\ 0 & 0 & 0 & 1 & 0 & 1 & 1 & 0 \\ 1 & 0 & 0 & 0 & 0 & 0 & 0 & 1 \end{pmatrix}. \tag{41}$$

The $T$, $C$ in Equation 34 and 35 are

$$T = \begin{pmatrix} \mathbf{0}_{8\times 8} & \mathbf{0}_{8\times 5} \\ \mathbf{0}_{5\times 8} & I_{5\times 5} \end{pmatrix}, C = \begin{pmatrix} I_{4\times 4} & \mathbf{0}_{4\times 9} \\ \mathbf{0}_{9\times 4} & \mathbf{0}_{9\times 9} \end{pmatrix}. \tag{42}$$

The permutation matrix $P \in \mathbb{R}^{13\times 13}$ rearranges the vertices according to the permutation $\pi$.

### A.2 Example of Satisfying Assignment and Permutation

### A.2.1 Satisfying Assignment 1

We assign **True** to: $x_1, x_2, x_3, x_4$, the truth assignment is: $x_1 = $ **True**, $x_2 = $ **True**, $x_3 = $ **True**, $x_4 = $ **True**. The permutation is:

$$\pi = [1, 3, 5, 7, 2, 4, 6, 8, 9, 10, 11, 12, 13].$$

This corresponds to a $P$ with:

$$P = \begin{pmatrix}
1 & 0 & 0 & 0 & 0 & 0 & 0 & 0 & 0 & 0 & 0 & 0 & 0 \\
0 & 0 & 1 & 0 & 0 & 0 & 0 & 0 & 0 & 0 & 0 & 0 & 0 \\
0 & 0 & 0 & 0 & 1 & 0 & 0 & 0 & 0 & 0 & 0 & 0 & 0 \\
0 & 0 & 0 & 0 & 0 & 0 & 1 & 0 & 0 & 0 & 0 & 0 & 0 \\
0 & 1 & 0 & 0 & 0 & 0 & 0 & 0 & 0 & 0 & 0 & 0 & 0 \\
0 & 0 & 0 & 1 & 0 & 0 & 0 & 0 & 0 & 0 & 0 & 0 & 0 \\
0 & 0 & 0 & 0 & 0 & 1 & 0 & 0 & 0 & 0 & 0 & 0 & 0 \\
0 & 0 & 0 & 0 & 0 & 0 & 0 & 1 & 0 & 0 & 0 & 0 & 0 \\
0 & 0 & 0 & 0 & 0 & 0 & 0 & 0 & 1 & 0 & 0 & 0 & 0 \\
0 & 0 & 0 & 0 & 0 & 0 & 0 & 0 & 0 & 1 & 0 & 0 & 0 \\
0 & 0 & 0 & 0 & 0 & 0 & 0 & 0 & 0 & 0 & 1 & 0 & 0 \\
0 & 0 & 0 & 0 & 0 & 0 & 0 & 0 & 0 & 0 & 0 & 1 & 0 \\
0 & 0 & 0 & 0 & 0 & 0 & 0 & 0 & 0 & 0 & 0 & 0 & 1
\end{pmatrix}, \tag{43}$$

and

$$PAP^T = \begin{pmatrix}
0 & 0 & 0 & 0 & 1 & 0 & 0 & 0 & 1 & 0 & 0 & 0 & 1 \\
0 & 0 & 0 & 0 & 0 & 1 & 0 & 0 & 1 & 0 & 1 & 0 & 0 \\
0 & 0 & 0 & 0 & 0 & 0 & 1 & 0 & 0 & 1 & 0 & 0 & 0 \\
0 & 0 & 0 & 0 & 0 & 0 & 0 & 1 & 0 & 1 & 0 & 1 & 0 \\
1 & 0 & 0 & 0 & 0 & 0 & 0 & 0 & 0 & 0 & 1 & 0 & 0 \\
0 & 1 & 0 & 0 & 0 & 0 & 0 & 0 & 0 & 0 & 0 & 1 & 0 \\
0 & 0 & 1 & 0 & 0 & 0 & 0 & 0 & 1 & 0 & 0 & 1 & 0 \\
0 & 0 & 0 & 1 & 0 & 0 & 0 & 0 & 0 & 0 & 1 & 0 & 1 \\
1 & 1 & 0 & 0 & 0 & 0 & 1 & 0 & 0 & 0 & 0 & 0 & 0 \\
0 & 0 & 1 & 1 & 1 & 0 & 0 & 0 & 0 & 0 & 0 & 0 & 0 \\
0 & 1 & 0 & 0 & 0 & 0 & 0 & 1 & 0 & 0 & 0 & 0 & 0 \\
0 & 0 & 0 & 1 & 0 & 1 & 1 & 0 & 0 & 0 & 0 & 0 & 0 \\
1 & 0 & 0 & 0 & 0 & 0 & 0 & 1 & 0 & 0 & 0 & 0 & 0
\end{pmatrix}, \tag{44}$$

we can easily verify that Equation 29 is satisfied.

$TPAP^TC$ is

$$
\begin{pmatrix}
0 & 0 & 0 & 0 & 0 & 0 & 0 & 0 & 0 & 0 & 0 & 0 & 0 \\
0 & 0 & 0 & 0 & 0 & 0 & 0 & 0 & 0 & 0 & 0 & 0 & 0 \\
0 & 0 & 0 & 0 & 0 & 0 & 0 & 0 & 0 & 0 & 0 & 0 & 0 \\
0 & 0 & 0 & 0 & 0 & 0 & 0 & 0 & 0 & 0 & 0 & 0 & 0 \\
0 & 0 & 0 & 0 & 0 & 0 & 0 & 0 & 0 & 0 & 0 & 0 & 0 \\
0 & 0 & 0 & 0 & 0 & 0 & 0 & 0 & 0 & 0 & 0 & 0 & 0 \\
0 & 0 & 0 & 0 & 0 & 0 & 0 & 0 & 0 & 0 & 0 & 0 & 0 \\
0 & 0 & 0 & 0 & 0 & 0 & 0 & 0 & 0 & 0 & 0 & 0 & 0 \\
1 & 1 & 0 & 0 & 0 & 0 & 0 & 0 & 0 & 0 & 0 & 0 & 0 \\
0 & 0 & 1 & 1 & 0 & 0 & 0 & 0 & 0 & 0 & 0 & 0 & 0 \\
0 & 1 & 0 & 0 & 0 & 0 & 0 & 0 & 0 & 0 & 0 & 0 & 0 \\
0 & 0 & 0 & 1 & 0 & 0 & 0 & 0 & 0 & 0 & 0 & 0 & 0 \\
1 & 0 & 0 & 0 & 0 & 0 & 0 & 0 & 0 & 0 & 0 & 0 & 0
\end{pmatrix}. \tag{45}
$$

We have $TPAP^TC\mathbf{1}_N = (0,0,0,0,0,0,0,0,2,2,1,1,1)^T$, which satisfies Equation 30. Here, the non-zero elements $(2,2,1,1,1)$ represent the number of literals evaluated to be **True** in each clause in Equation 36.

### A.2.2   Satisfying Assignment 2

We assign **True** to: $x_1, x_2, \neg x_3, x_4$ with truth assignment $x_1 = $ **True**, $x_2 = $ **True**, $x_3 = $ **False**, $x_4 = $ **True**. The permutation is:
$$\pi = [1, 3, 6, 7, 2, 4, 5, 8, 9, 10, 11, 12, 13].$$

### A.2.3   Satisfying Assignment 3

We assign **True** to: $x_1, \neg x_2, x_3, \neg x_4$ with truth assignment $x_1 = $ **True**, $x_2 = $ **False**, $x_3 = $ **True**, $x_4 = $ **False**. The permutation is:
$$\pi = [1, 4, 5, 8, 2, 3, 6, 7, 9, 10, 11, 12, 13].$$

### A.2.4   Satisfying Assignment 4

We assign **True** to: $\neg x_1, x_2, \neg x_3, \neg x_4$ with truth assignment $x_1 = $ **False**, $x_2 = $ **True**, $x_3 = $ **False**, $x_4 = $ **False**. The permutation is:
$$\pi = [2, 3, 6, 8, 1, 4, 5, 7, 9, 10, 11, 12, 13].$$

### A.2.5   Satisfying Assignment 5

We assign **True** to: $\neg x_1, \neg x_2, \neg x_3, \neg x_4$ with truth assignment $x_1 = $ **False**, $x_2 = $ **False**, $x_3 = $ **False**, $x_4 = $ **False**. The permutation is:
$$\pi = [2, 4, 6, 8, 1, 3, 5, 7, 9, 10, 11, 12, 13].$$

