# OpenReview forum: "Permutation Picture of Graph Combinatorial Optimization Problems"
_TMLR — Rejected by TMLR_

### Review · Reviewer_oRkR · 2024-11-12

**Summary Of Contributions:**

The bulk of the paper is Section 4 on "Formulating Graph Combinatorial
Problems using Permutation" where 9 graph combinatorial problems are
reformulated in such a way that a permutation forms part of the
problem (either in the objective or in a constraint). One of these
problems is SAT which is encoded graphically.

The abstract states that: "This paper proposes a framework that
formulates a wide range of graph combinatorial optimization problems
using permutation-based representations" but there is no framework in
the sense of a general-purpose method for permutation-based
reformulations. Instead we get 9 separate reformulations, albeit with
some common themes.

The motivation for these reformulations is to "build surrogate losses
using permutations" with a view to using these surrogate loss
functions in an unsupervised learning approach (to solving the
original problem) where the surrogate loss acts as a proxy for the
original optimisation objective. The authors argue that a surrogate
loss based on permutations is better that one based on QUBO. The main
argument offered against QUBO is its lack of intuitiveness for example
for TSP. Since TSP is essentially learning a permutation (of cities to
visit) a permutation-based method is indeed more "natural" (whether it
leads to faster solving is another matter). However, the main
argument is that "our approach can be combined with recent advancements
in differentiable sorting and ranking techniques, opening up
possibilities for applying gradient-based (differentiable) machine
learning methods to these discrete optimization problems". Ultimately,
the goal is to use first-order methods on a continuous reformulation
of the original problem. Methods developed for ML can be applied here,
although we are not actually doing ML since we are using a surrogate
loss not a loss computed from data.

The writing is clear and I did not find errors in the reformulations
so the key issue is whether the contribution here is sufficient for
publication in TMLR. I think not - the paper is too preliminary. We
merely have the suggestion that these reformulations can lead to
useful gradient-based ML approaches to solving the given problems. I
think we need to see this combination effected and empirically
evaluated to merit publication. Note that ultimately we are interested
in how the promised approach would compare to existing methods for
solving these problems. For TSP it would be interesting to compare
with the Concorde solver, for example. Of course, nothing close to
that is offered here.

The authors state that "Permutations [offer] intuitive and efficient
formulations that often align closely with problem structures." I
think the word "often" here is not useful (how often?). The fact is
that permutations make sense where the problem is essentially about
ordering and a poor choice when this is not the case. For example for
independent set, in the authors' formulation, all that matters is that
the first k vertices in the ordering/permutation form an independent
set - how these are ordered and how the remainder are ordered does not
matter, so we will get a lot of (undesired) symmetry.

**Audience:**

Yes

**Claims And Evidence:**

Yes

**Requested Changes:**

As stated in the main review "Summary of Contributions" the promised application of these reformulations in a surrogate-loss approach to the original problems is needed. In addition a comparison to SOTA methods for these problems should be there.

**Strengths And Weaknesses:**

See above. The main contribution is the 9 reformulations. The main weakness is that offering these 9 reformulations is not enough to merit publication.

---

### Review · Reviewer_jrgX · 2024-11-12

**Summary Of Contributions:**

This work considers combinatorial optimization problems in a fairly broad sense. It proposes that they can be approached by thinking about permutations over the graph as a fundamental concept and the search space in which algorithms should operate. The authors go through a series of combinatorial optimization problems (such as TSP, MIS, MaxCut, ...) and arrive at a series of formulae for representing them in this way. High-level proofs are given for each problem transformation. It is argued that this opens the door to a new family of combinatorial optimization approaches using unsupervised learning. This would inherently need to rely on the existence of techniques for backpropagating through discrete distributions in order to make gradient descent approaches feasible.

**Audience:**

Yes

**Claims And Evidence:**

No

**Requested Changes:**

Please address W1 and W2 above. Additional comments:

C1. The proofs are hand-wavy and can be difficult to follow, though they appear correct. A more formal treatment would be preferable in my opinion.

C2. I'd recommend moving the content in Section 5 right before all the formulations, since they outline the strategy for constructing the correspondences.

C3. I'd suggest adding a "Notation / Preliminaries" section since many fundamental concepts are defined over and over, such as: the number of nodes, adjacency matrix, permutation matrix, etc. They can be specified once and assumed known for the rest of the paper. The derivations will be more succint.

C4. Some of the formulae do not typecheck -- take Equation 8, the objective should be to maximize $k$ subject to the constraint, maximizing an equality relation does not make sense. I would also recommend keeping either the permutation matrix $P$ or the permutation $\pi$ as the argument for the optimization, both are used inconsistently in the paper.

C5. Check use of  `\citep` and `\citet` throughout the paper (e.g., Section 1.3)

C6. 4.3.3: $T$ already denotes transpositions, choose another symbol

**Strengths And Weaknesses:**

### Strengths

S1. The work provides an interesting and arguably novel perspective on these problems. I tend to agree with the authors' opinion that permutations are more natural for sequencing and ordering problems, as opposed to learning independent quantities, as has been done in recent unsupervised ML work for CO. The fact that the framework could be quite general is supported by the derivations.

### Weaknesses

W1. The main weakness of the paper is that it does not contain any evaluation. While the basic premise indeed has potential, the paper stops short of carrying out any experiments. Currently, the paper reads more like the description of a research programme or an agenda for a number of years, more than a publishable unit of work. More practical evidence (or indeed *any* evidence) is needed, in my opinion, to support the key hypothesis.

W2. Following on from the previous point, I believe that there will be thorny technical details associated with implementing this in practice for each problem.
- A relevant question is how will you deal with problems containing constraints (MIS, graph coloring, MVC, ...) -- how can you ensure that they are satisfied when using continuous optimization? You could start by evaluating those that do not require constraints (TSP, MaxCut, ...).
- There is also the potential issue that the presented transformations increase the computational cost of evaluating a solution. The TSP, for example, grows in cost from $\mathcal{O}(m)$ (just add up the costs of the edges) to nominally $\mathcal{O}(n^3)$ (assuming naive matrix multiplication).  It is unclear if this trade-off would be practical.

W3. The presentation is somewhat deficient (redundant / vague / sloppy) in places. It should be improved, especially as there are no experiments.

---

### Review · Reviewer_HrZB · 2025-01-21

**Summary Of Contributions:**

The paper proposes a framework to encode graph-combinatorial problems into Unsupervised Learning problems by constructing a surrogate loss through permutations.
The approach is proposed to overcome the binary constraint of QUBO formulation. The paper then continues by proving the applicability of the proposed framework to several well-established graph-combinatorial problems.

**Audience:**

Yes

**Claims And Evidence:**

No

**Requested Changes:**

- Improve writing in *Introduction* Section
- Include a discussion about the complexity of re-order w.r.t the dimension of the search space.
- Either test the framework using ML algorithms over the chosen graph-combinatorial problems **or** include a more detailed explanation about how this framework can be linked to state-of-the-art machine learning methods (I refer to the second paragraph of  *Conclusion* section)
- (Optional, but would strengthen the paper), add an example as the one used in the *Discussion* Section, for each graph-combinatorial problem used in Section 4.

**Strengths And Weaknesses:**

**Strengths**
- The proposed framework is interesting since it aims to resolve the binary limitations of QUBO.
- The theoretical part (Section 4), is well done.
- The idea of using permutation (hence, re-ordering) is smooth and original.

**Weaknesses**
- The framework is presented only in a theoretical manner, no ML algorithms have been tested on the selected graph-combinatorial problems to test the effectiveness of the proposed framework.
- Too much is left for future work or future direction with no details explanation about how to connect well-known ML algorithms to such ML4CO formulation.
- No discussion about the complexity required by the re-ordering w.r.t. problem search space.

---

### Decision · Action_Editor_yQM3 · 2025-02-14

**Recommendation:** Reject

**Comment:**

There is a clear consensus among the reviewers that the paper is not ready for publication, due to issues about clarity and presentation but most importantly due to a lack of experiments to support the presented claims.

All reviewers acknowledge the potential of the presented ideas, and encourage the authors to strengthen their results with algorithmic and experimental results to demonstrate its benefits.

**Audience:**

yes

**Claims And Evidence:**

no - there is a clear consensus among the reviewers that the paper is not ready for publication, due to issues about clarity and presentation but most importantly due to a lack of experiments to support the presented claims.